# Indoor noise level measurements and subjective comfort: Feasibility of smartphone-based participatory experiments

**Carlo Andrea Rozzi**[1]*, **Francesco Frigerio**[2], **Luca Balletti**[3], **Silvia Mattoni**[3], **Daniele Grasso**[4], **Jacopo Fogola**[4]

**1** Istituto Nanoscienze—Consiglio Nazionale delle Ricerche, Modena, Italy, **2** ICS Maugeri Spa—Centro Ricerche Ambientali, Pavia, Italy, **3** Unità Comunicazione e Relazioni con il Pubblico—Consiglio Nazionale delle Ricerche, Roma, Italy, **4** ARPA Piemonte, Torino, Italy

* carloandrea.rozzi@nano.cnr.it

**Data Availability Statement:** We made the experiment data publicly available via the Open Science Foundation at the URL https://osf.io/nufmx.

## Abstract

We designed and performed a participatory sensing initiative to explore the reliability and effectiveness of a distributed network of citizen-operated smartphones in evaluating the impact of environmental noise in residential areas. We asked participants to evaluate the comfort of their home environment in different situations and at different times, to select the most and least comfortable states and to measure noise levels with their smartphones. We then correlated comfort ratings with noise measurements and additional contextual information provided by participants. We discuss how to strengthen methods and procedures, particularly regarding the calibration of the devices, in order to make similar citizen-science efforts effective at monitoring environmental noise and planning long-term solutions to human well-being.

## Introduction

Monitoring acoustic environments requires substantial dedicated human and material resources. Assembling a large number of densely sampled observations over the whole national territory is a difficult and expensive task. In fact, a single institution can acquire the hardware and perform proper calibrations up to few class 1 measuring devices. The use of smartphones has been proposed as a cost-effective method for assembling networks of environmental noise detectors.

In the last decade several participatory sensing initiatives were launched to explore the possibility of exploiting the capillary diffusion of digital technologies and to empower citizens to collaborate with scientists. *NoiseTube* was the first acquisition app developed together with a geo-localization platform meant for passive systematic monitoring to report daily noise data acquired by the users [1]. Several other apps and initiatives followed. The *NoiseSpy* app [2] allows collaborative collection and visualization of urban noise levels in real-time. The *City-Sense* [3] and *Beyond The Noise* [4] projects moved steps forward to a more comprehensive definition of soundscape and acoustic comfort in public places and the identification of quiet areas in an urban context. *NoisePlanet* (http://noise-planet.org) aims at collective sensing and

**Funding:** The authors received no specific funding for this work.

**Competing interests:** The authors have declared that no competing interests exist.

modeling through a community of contributors. Data and maps collected with *NoiseCapture* were made publicly available [5]. Large amounts of geolocalized data were also collected worldwide within *EveryAware* (http://www.everyaware.eu), a project meant to collect citizen-science initiatives in several different environmental fields. For extensive reviews on the topic see [6, 7].

The reports of those experiences show that participatory sensing campaigns are prone to weaknesses that must be taken into consideration and minimized [8, 9]. First, app retention is known to be low (the majority of participants performs at most one or two measurements), therefore projects relying on long term participation are unlikely to succeed. In addition, the lighter and easier to install the app is, the greater is the probability of download and use. Second, while most participants just perform very few measurements, a minority of them contribute a large number of measurements, i.e. there is a serious risk of biasing the results. Third, the hardware-software reliability is not trivial to keep under control, given the fast-paced development of smartphones, and the increasing variety of producers on the market. Last, performing environmental noise measurements is a deceptively simple task, also due to the large number of inaccurate "dB measurement apps" available [10]. Untrained participants may turn into a huge source of meaningless data if not properly instructed.

We designed a participatory-sensing campaign aimed at checking the feasibility of correlating subjective comfort ratings with noise levels measured by participants in such a way to minimize the impact of known issues.

The campaign had a limited time span, and all of the measurements were taken from fixed locations (participants' homes). Therefore we focused on measuring indoor noise affecting residential areas only. Our protocol does not involve geolocalization, as tracking participants at home might have discouraged many people from employing the app. This feature completely disentangles participants' sensitive information from the activity, and guarantees complete anonymity. Giving up geolocalization has an impact on the spatial resolution of the data. However, our initiative did not aim at compiling noise maps, a goal that would require a much more detailed description of the locations. Fortunately, the median size of Italian municipalities is very low (around 2500 inhabitants), and the (few) large urban territories are partitioned by postal codes, therefore our resolution is the minimum territorial partition provided by either identifier. Usually it corresponds to the municipality territory. In our protocol neither the values, nor the location and subjective data are uploaded to a central server by the app.

A detailed fact sheet was made available including a step-by-step guide for taking measurements and sending data and a series of specific insights to contextualize the experiment (by explaining what sound is, how it is measured, what noise pollution is, what can be measured with smartphones and what the limits of the instrument are).

Participants were left free to make as many measurements as they wished at home to grasp the general features and dynamics of their local acoustical environment, but they were finally requested to send only two records representing extreme situations (i.e. the noisiest and the quietest in their environment). This kind of sampling is, above all, dictated by the need to keep the procedures simple and the participants' burden minimal. It does not provide a continuous-time representation of the acoustical comfort at home, still it allows us to extract enough information about the relationship between perceived comfort and measured noise levels.

A smartphone can work as a Sound Level Meter (SLM), provided the proper software is installed, with inherent limitations due to the fact that the quantities of interest, such as LAeq, are obtained by computation from a sampling of the audio input. A common issue to all mobile crowd-sensing initiatives is that data collected by many users employing random devices and running different softwares might turn out to be nearly useless due to poor reliability [11].

Several authors investigated whether smartphones can be reliably employed as sound level meters. Kardous and Shaw [12, 13] assessed the performance of 10 iOS and 4 Android apps, and found that, while some apps are inaccurate, some of them are appropriate for occupational noise measurements. They report that the smallest mean difference for A-weighted, uncalibrated data is within ±2 *dB(A)* for the *NoiSee* and *SoundMeter* apps, when measuring pink noise with frequency range 20 *Hz*—20 *kHz* and noise level 65–95 *dB* in a lab. Accuracy is increased to ±1 *dB(A)* when external microphones are employed. Murphy and King [14] found that iOS platforms are superior to Android ones in a reverberation room test. Ventura et al. [15] reported a measurement error standard deviation below 1.2 *dB(A)* within a wide range of noise levels. Celestina et al. [16] showed that a sound level meter app and an external microphone can achieve compliance with most of the requirements for Class 2 of IEC 61672/ ANSI S1.4–2014 standard.

The reliability of smartphones and related apps as sound measurement tools out of the lab depends on a variety of factors such as quality of hardware and software, operating system, age and condition and measurement procedures on the field. The calibration of professional SLMs is checked on the field by fitting the microphone to a calibration source. Indeed, smartphones are not designed for metering. They are optimized to output voice and possibly music in the best cost-effective way [17]. With the use of external microphones, a limited number of calibrated detectors equipped with effective logging software can be set up [18]. However, plugging an external microphone in the analog input is not practical for everyday users. A method has been proposed [19] to calibrate the internal microphone using environmental noise as a source, but it requires a calibrated microphone as well, and requires sampling a certain amount of data in order to achieve proper averaging of the spectral response.

In our protocol we choose the open source SLM software *OpeNoise*, developed by one of the authors [18, 20]. The app was not designed, at first, for participatory sensing, but with the aim at setting up low cost noise detector networks, thus it does not support geolocalization or automatic data uploading to a dedicated server and the first testing was performed with external microphones. However it has some features useful for educational purposes such as ⅓ octave and FFT real-time display that make it suitable for high-school students.

The use of a single app ensures that the same noise signal processing algorithm is used by each participant; the differences due to hardware are thoroughly examined in [12, 13]. Our tests [17, 18] showed that, when the dynamic range and spectral content are within the typical values of urban noise, below 85 *dB(A)* and with no relevant pure tones around 6–8 kHz, different smartphone models behave in a very similar way.

We focused in the first place on comparing different comfort states, by looking at differential measurements, which are less affected by calibration issues. However the app itself allows to adjust a device-dependent single parameter gain; in post processing, a subset of data was corrected for the gain, when known. In this way data can be examined in "dual mode", which offers the advantage of allowing cross-validation to some extent (see discussion below).

Citizen-science initiatives offer several societal and educational benefits [21]. In our case we mention: impacting citizens' perception of the value of scientific data by directly involving them in a scientific research project; contributing to education in STEM area by providing basic information about sound and its measurement and making people aware of the difficulties in making reliable measurements; spreading knowledge of the merits and limits of measuring sound with smartphones; contributing to alleviate negative effects of COVID-19 lockdown, especially on high-school students. Moreover, the involvement of pupils and teachers belonging to the compulsory education compartment is fundamental in ensuring equal opportunities for inclusion of people from every social status, and, by the way, it avoids that the sample is biased, e.g., by people owning the most expensive smartphones, or living in

privileged contexts. Last, but not least, the possibility of empowering citizens to take an active part in environmental issues, potentially affecting their health, is in line with EU directive 2003/35/EC.

## Materials and methods

The experiment was first proposed in Bergamo, one of the provinces most affected by the first wave of COVID-19 in Italy. In collaboration with the BergamoScienza science festival, between October 12 and 14 2020 the experiment was presented to a dozen schools with a live broadcast in which we explained the purpose of the experiment and provided instructions to participate. The feedback obtained from the first measurements was useful to fine tune the protocols before the national launch, which took place on October 19 with a live broadcast within the program of the National Geographics Science Festival in Rome and on the social channels of the Communication and Public Relations Unit of the Cnr. Data were collected from Oct 19 until Nov 15, during this period several social relaunches were made to involve more people to participate. About a month after the end of the experiment, a new live broadcast was organized to share some preliminary results with the participants. Further details about the communication actions are provided in the S1 File.

The instructions, additional information and didactical materials for teachers and students were distributed through a dedicated website [22], where a Google form was made available to send the collected data. The terms and conditions of use of the form clearly stated that the data were collected and analyzed anonymously and that compiling and sending the form represented the participant's written informed consent to use the data for research purposes only, as explained in the information material available on site. An ethical committee approval was not sought because no personal or sensitive information was collected within the study. Furthermore the OpeNoise privacy statement clarifies that the app does not record or share any personal data, it does not record audio and does not acquire information about the localization of the smartphone. On the same page a browsable world map displayed all the collected measurements in real time, allowing people to navigate raw data. Particular care was put in the instructional material to stress factors that could render the measurements invalid. It was clearly explained how to handle the smartphone with extreme care, to properly identify and direct the microphone, to be aware of interfering noises both from the inside and the outside of the house, and it was recommended to reset and repeat the measurement in case something went wrong.

Performing the experiment required a smartphone, paper and pen, installing the OpeNoise app and following the instructions articulated in three phases (see S1 Text in S1 File for the detailed protocol): **Phase I** (preparation and awareness): familiarizing with the app and becoming aware of the acoustical background in participants' homes; **Phase II** (colloquially named "measuring noise"): rating subjectively the acoustical comfort level and performing indoor noise level measurements from an open window. Then select two records corresponding to the subjective quietest and noisiest states experienced and send them via the online form. **Phase III** (optional, colloquially named "measuring silence"): perform an indoor measurement of the quietest background with all windows shut.

Considering the difficulties in performing an accurate calibration on the field we asked participants *not* to apply gain corrections to the *OpeNoise* app. The optional background measurement (Phase III) was performed by about half of the participants. This fact allowed us to check the reliability of correcting *ex post* the smartphone gains, based on a previous calibration campaign performed in a lab controlled environment (see the discussion below).

In defining our measurement protocol we took into account Italian law regulating measurements of noise pollution [23] which requires positioning at a distance of 1 *m* from open

windows to measure disturbing noises coming from the outside. Participants were instructed to suppress every indoor source of disturbance and to repeat the measurement in case of unexpected disturbance.

Residual noise (i.e. indoor background noise) can be cancelled during the analysis stage by subtracting levels measured in the quiet situation (i.e. without disturbing external noises) from the levels measured in the noisy situation (i.e. with disturbing external sources).

## Results

A total of 492 records containing 1258 measurements were received. 91% of the records were compiled by high-school students 14 to 18 years old.

Participants contributed data from 12 Italian regions (out of 20). 59% of the records originated from two northern regions (Emilia-Romagna and Lombardia) whose combined population is 24% of the total Italian population. The median population of the municipalities in which the participants declared to live was 17198. The largest cities contributing were Torino (over 887000 inhabitants, 46 records received) and Genova (about 586000 inhabitants, 43 records received). 49% of the participants declared to live in an area with high population density, 42% in an area with low population density, and 9% in isolated premises (e.g. rural or mountain areas, etc.). Overall 73% of the measurements were performed from windows overlooking places open to the public such as public streets, squares, etc.; the remaining 27% overlooked inner spaces such as private backyards or gardens. 81% of the measurements were performed within the height of the second floor. 36% of the smartphones employed were made by Apple running iOS, the remaining 64% were from mixed brands running Android (33% Samsung, 16% Huawei, 10% Xiaomi, etc.). These figures approximately reflect the Italian smartphone market shares [24].

Road traffic noise was the most frequently cited source of noise overall (33%). However the frequency of occurrence is different in the subsets of measurements referring to noisy and quiet states. In quiet situations natural sounds are the most frequent (35%) followed by road traffic (27%), unidentified (14%) and outdoor anthropogenic (12%), while in noisy situations road noises are most frequently identified as the main source (40%), followed by outdoor anthropogenic (21%), natural (13%) and inner anthropogenic (11%). Other sources are mentioned in less than 6% of the records in both situations. The ratio of traffic/industrial to natural/undefined sources decreases from 5:1 in noisy states to almost 1:1 in quiet states (see S1 Fig in S1 File).

The records were first carefully screened for major inconsistencies. 33 records were discarded because they contained duplicated data or impossible measurements (such as $LAmin > LAmax$, $LAeq < LAmin$, etc.). 14 records reported lower comfort ratings for the quiet state than for the noisy state, together with lower noise levels in the quiet state. These records are not necessarily wrong as a louder environment could be judged more comfortable than a quieter one under some circumstances, but not enough information was provided to investigate the reasons for these anomalies, so they were removed. In 65 cases the upload order appeared to be inverted (i.e. data reporting both lower comfort rating *and* higher LAeq were uploaded to the web page entitled "quiet state" and vice versa). While removing completely these records from the dataset does not substantially change the mean values of comfort and LAeq (see S1 Table in S1 File) we decided to swap the labels and keep these records as they contain valid and consistent data. This glitch shows that exploiting redundant information is important for data sanity-checks and to monitor the reliability of the data flow. 16 records reported a suspiciously small time interval (< = 1 min) between the recording of the noisy and quiet states. However, since the inclusion of these records did not subvert the conclusions of

this work, we choose to keep them in the mix. An independent set of consistency checks was performed on the sound level values in both Phase II and Phase III (see discussion below). In sum 93% of the records were considered valid, which demonstrates a remarkable collaboration effort by participants.

Comfort was rated on a scale from 1 (less comfortable) to 5 (more comfortable). The distribution of the overall ratings (Fig 1) is skewed towards high values (i.e. high comfort). 55% of the records rate comfort $>= 4$, 25% comfort $= 3$, and 20% comfort $<= 2$. The mean rating for

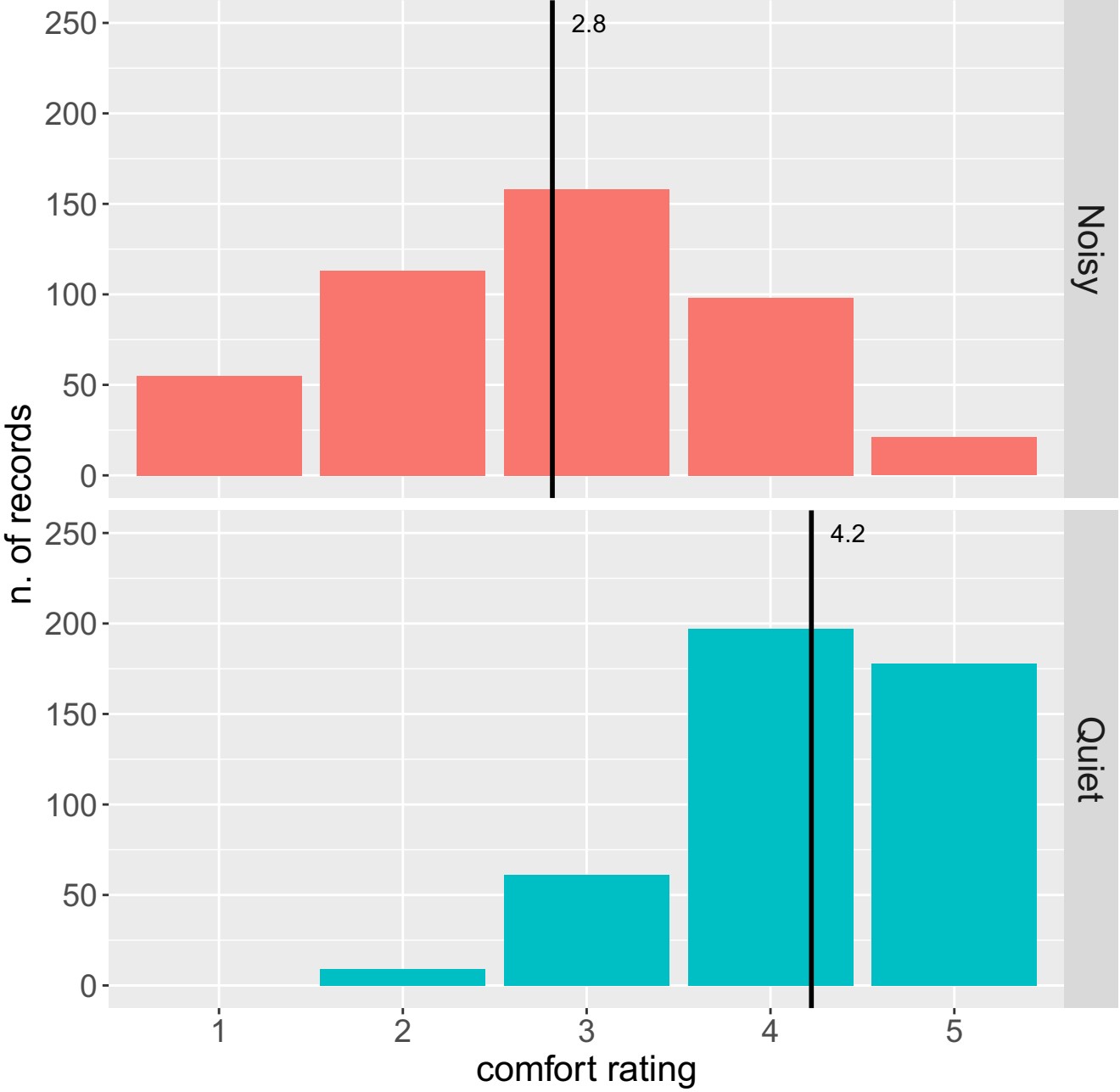

**Fig 1. Number of comfort ratings by comfort state.** Black lines indicate the mean values. The mean comfort score for quiet states ($N = 445$, $M = 4.2$, $SD = 0.8$) is higher than that for noisy states ($N = 445$, $M = 2.8$, $SD = 1.1$).

quiet states is 4.2, and 72% of the ratings are $> = 4$. The mean rating for noisy states is 2.8 and 30% of the ratings are $< = 2$. The mean comfort score during daytime (3.4) is lower than at night (4.1) (see S2 Fig in S1 File).

Raw (uncalibrated) data can be analyzed as differences between noise levels measured with the same smartphone. Since each participant contributed two measurements per record sent (corresponding to the noisiest and quietest state at his/her location) we can study level differences neglecting calibration issues at first, assuming that the sources do not induce strong frequency-dependent nonlinearities in the responses. This analysis is also important to rule out the effect of residual background noise at each location.

The distribution of $\Delta LAeq = LAeq(quiet)—LAeq(noisy)$ for different values of $\Delta comfort = comfort(quiet)—comfort(noisy)$ quantifies the change in the perceived comfort between the two extreme states in each observer's home (see Fig 2). When there is no difference in the perceived comfort the median $\Delta LAeq$ is correctly found to be not significantly different from 0 $dB(A)$. This is a very good indicator of data consistency.

**A linear model shows that $\Delta LAeq$ decreases with increasing difference in the perceived comfort at a rate of 5.9 $dB(A)$ per point of comfort loss (95% CI: [-6.7, -5.1] $dB(A)$).** This figure can be compared with the smaller dataset of calibrated values reported below as a validation that applied gains are correct.

The measurement of the quietest background noise experienced indoors (in the quietest room, with windows shut) was proposed with a twofold purpose: on one hand to explore the possibility of applying a simple calibration scheme prepared beforehand; on the other hand to identify possible unexpected behavior in some smartphones.

Model-dependent additive constant gain corrections were applied to raw LA values (see S2 Table in S1 File). The corrections were obtained by comparing noise levels measured by the smartphone and a Class-1 sound level meter, using a pink noise source with sound levels in the range from 50 to 80 $dB(A)$. Each value reported per producer/model is referred to a single smartphone tested; the variability in the gain response within models was not investigated, except for iOS ones. Remarkably, it turns out that gain corrections for different iOS models agree within ±1 dB probably thanks to a more standardized manufacturing process. The ratio of iOS- to Android-based models in the calibrated database differs from the ratio of the whole sample. The fraction of iOS phones in the calibrated database is 72%. Therefore we expect that the calibrated dataset is overall both more precise and more accurate than the complete dataset.

**Calibrated background noise data (Fig 3) show a marked difference in the reliability at low noise levels between iOS and Android based smartphones.** The minimum of the distribution of values for iOS-based smartphones is 26 $dB(A)$, with a very sharp mode at 28.6 $dB(A)$. Android smartphones behaved differently. Median and dispersion of Samsung smartphones (Md = 33.4, $IQR = 16.2\ dB(A)$) are comparable to the ones of Apple smartphones (Md = 32.6, IQR = $12.9\ dB(A)$), even if 6 values are smaller than the Apple minimum. This indicates that the applied gain is probably incorrect for these models. The misalignment could be either due to a mistake in the model classification (some models come in several versions sharing the main name and differentiating by a suffix such as "Pro", "Extra", "Plus", etc., which were neglected) or to a dependence of the gain on the specific version of the operating system (which was not tracked). Five records employing non-Samsung smartphones reported values $< 20\ dB(A)$ or even negative values after calibration. Either these are cases of mistyped values, or these smartphones are out of their linear operating range, and are unable to capture the noise level of an extremely quiet environment regardless of the gain correction. The measurements performed with these models were considered unreliable and discarded.

We applied calibration also to ordinary (non-background) measurements of Phase II. After calibration fewer extreme noise levels were left, being worth a closer look. Nine LAeq values

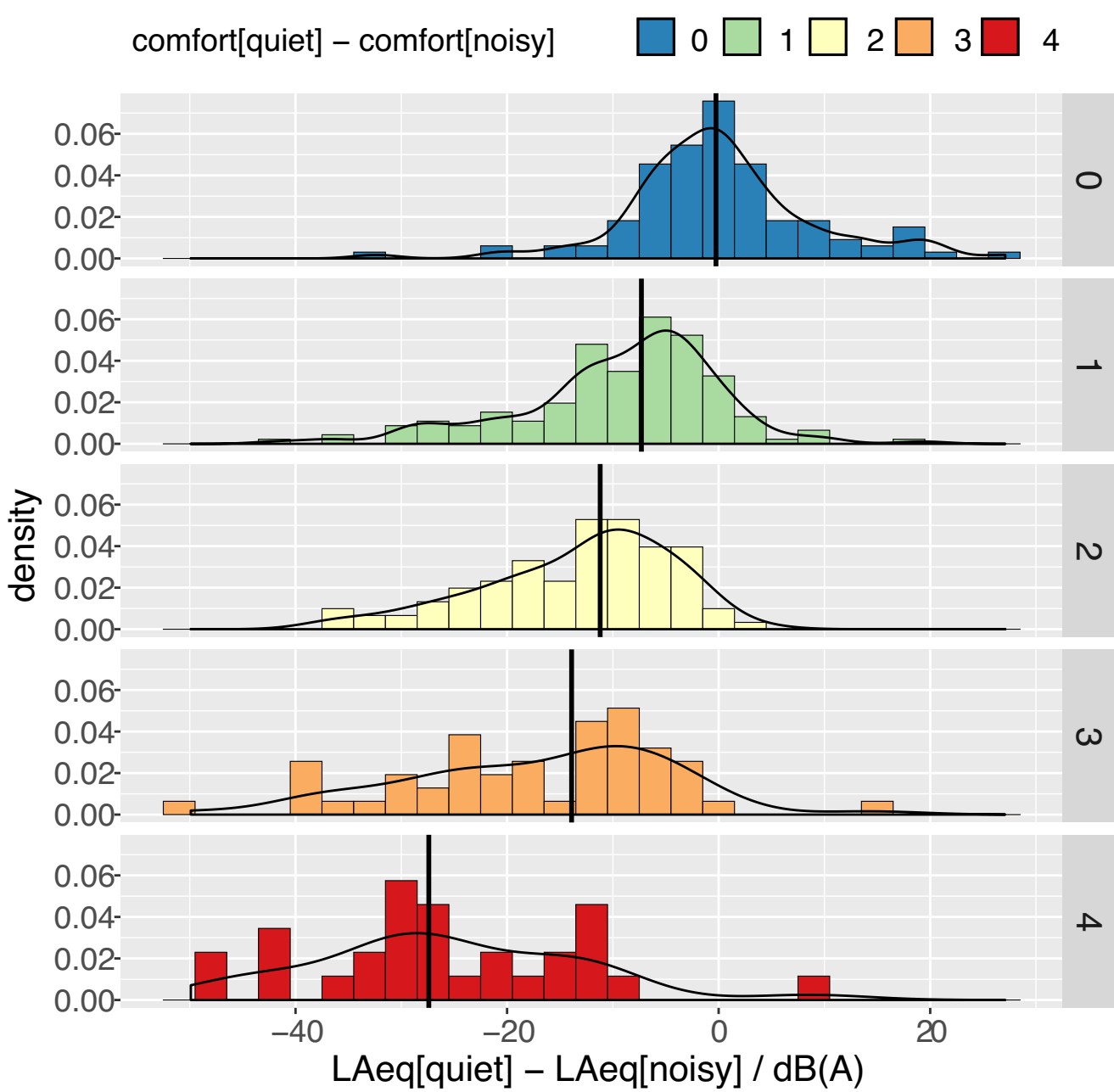

**Fig 2. Distribution of ΔLAeq = LAeq[quiet]—LAeq[noisy] for different values of Δcomfort (difference of subjective comfort ratings between quiet and noisy states felt by each participant).** Black lines indicate median values. A linear model yields $\Delta LAeq/dB\ (A) = -1.54 - 5.91 \times \Delta comfort$ ($F_{(1,443)} = 214$, $p < 0.001$, $R^2 = 0.325$).

were below 25 *dB(A)*, which are incompatible with the results of Phase III, as we have seen that the lowest possible threshold in this dataset is 26 *dB(A)*, considering sufficiently reliable the results obtained from iOS-based smartphones (see also [14]). All of them refer to the same smartphone model, a circumstance that suggests that this is probably a case of wrong model classification, for which the applied calibration is just not appropriate. These values were removed from the dataset.

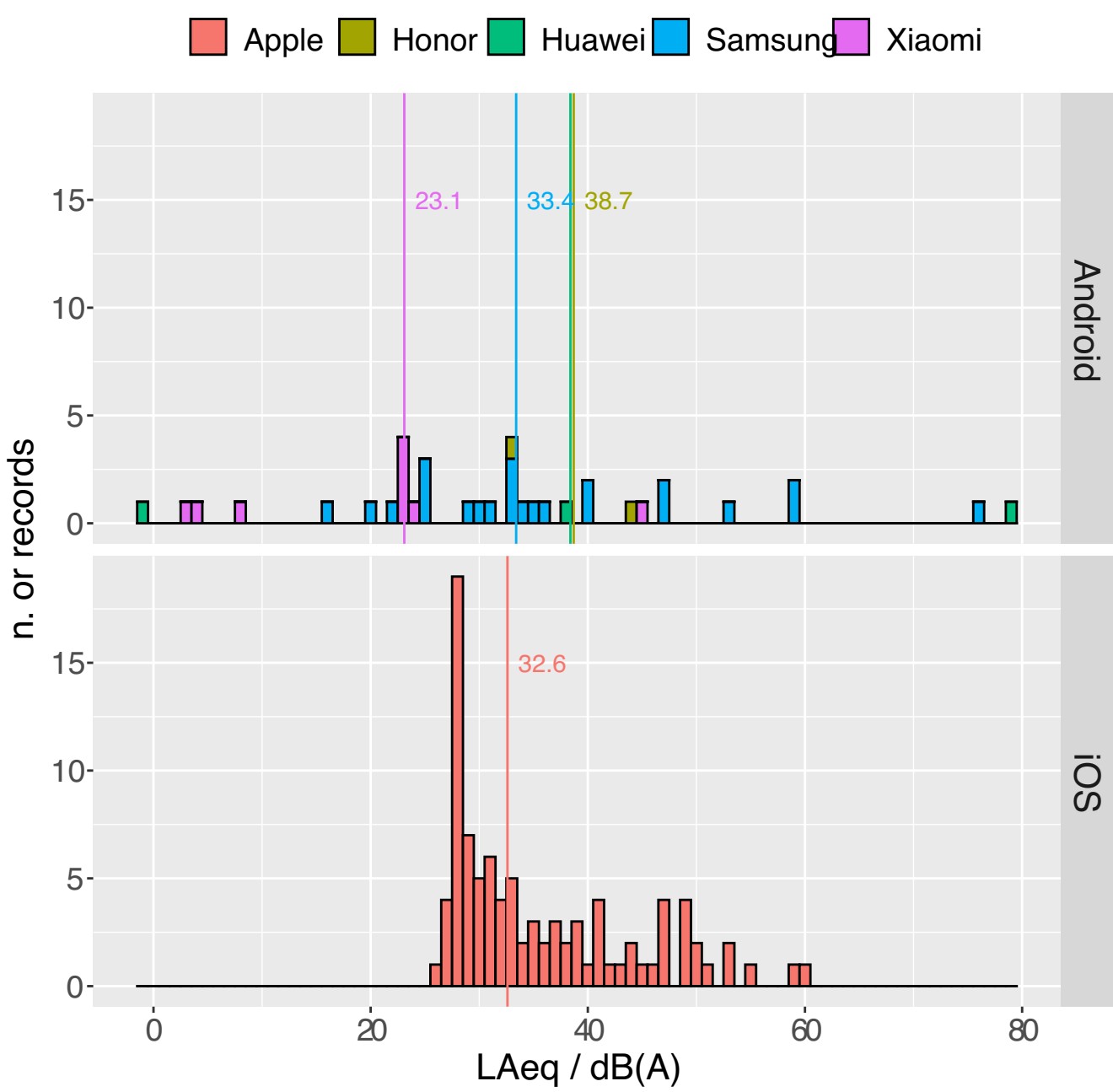

**Fig 3. Dispersion of calibrated LAeq for the quietest background.** The values were obtained by adding to the raw LAeq the gains reported in S2 Table in S1 File.

In 19 cases LAeq was > 75 *dB(A)*, with a median LAmax of 93 *dB(A)*. In some of them participants complemented the measurement with comments that justify such high values. For example: LAeq = 76.6 *dB(A)*: "heavy property renovation in the neighbor's apartment"; 76.7: "ambulance"; 77.7: "leaf blowing during road cleaning"; 79.7: "heavy traffic on the motorway"; 80.0: "car accident". It is clear that these readings describe specific events. Some of them appear to be accidental, and are not particularly representative of the average acoustic environment, but in some other cases they might be recurring, thereby substantially contributing to the

acoustical perceived wellness in the area where they were observed. In this experiment these cases cannot be further distinguished.

After applying calibration and removing unphysical readings below 26 *dB(A)* a total of 436 measurements were left to be analyzed. Fig 4 shows the distribution of LAeq attributed to quiet and noisy states. **The median gain-corrected LAeq are 49.1 *dB(A)* in noisy and 41.5 *dB (A)* quiet situations.**

The asymmetry of the distributions in Fig 4 requires a comment. The shape for noisy states shows a heavier tail towards high LAeq, as expected, while the distribution for quiet states

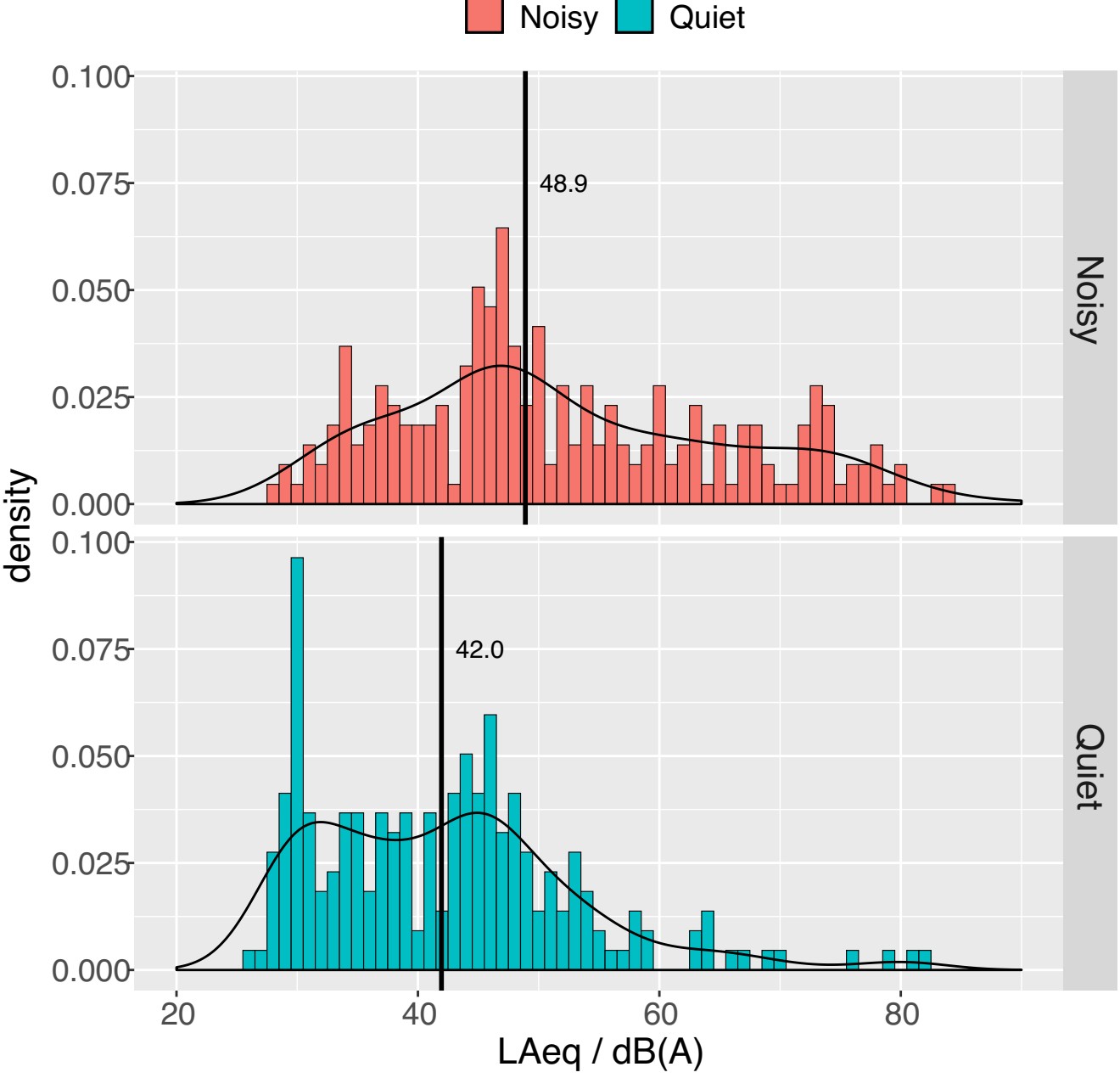

**Fig 4. Distribution of LAeq for noisy and quiet states.** Thick black lines show median values. A few outliers are visible on the right side of the quiet state distribution. The outliers below 26 dB were considered due to inappropriate calibration and were removed from the dataset.

(aside from a few outliers, which represent sudden events), is clearly bimodal. Besides the peak in common with the noisy distribution around 46 dB(A), another dominant feature appears at 30 *dB(A)*. Decomposing the plots further into their night and day projections clarifies the origin of this peculiar shape.

We have shown above that a direct relationship exists between differential noise levels and differences in comfort ratings. By studying calibrated values we can also show that a similar association exists between absolute LAeq and absolute comfort ratings.

Fig 5 shows that the median LAeq decreases monotonically as a function of the comfort rating. The extreme values of the median LAeq are 61.5 *dB(A)* for the lowest comfort score, and 39.1 *dB(A)* for the highest comfort score. **The association between measured LAeq and the absolute comfort rating is statistically significant ($F_{(1,434)}$ = 126, $p < 0.001$, $R^2$ = 0.24) under the linear relation *LAeq / dB(A) = 67.1–5.6 × comfort*, with 95% CI for the slope coefficient: [-6.5, -4.7] *dB(A)/comfort point*.** This figure is very close to the value obtained to describe the variation of LAeq as a function of the variation of comfort, confirming that the trend is robust with respect to calibration. Obviously this fit is not meant to be predictive of individual values, which are also associated with other factors and to clustering of the observations. It rather describes an average trend in the whole dataset. The slope for night data is comparable to the one for day data: 95% CI: day [-6.3, -4.3]; night [-7.5, -2.3].

The data show an overall clear trend despite considerable variability. In a participatory experiment (as opposed to a laboratory controlled environment) many factors contribute to this variability. Among them the fact that sources have an intrinsic dispersion in acoustical features, unknown variation in smartphone configuration and frequency-dependent microphone sensitivity (i.e. beyond the one accounted for in the calibration procedure we performed), individual errors due to failure to strictly follow the measurement protocol, etc. We can provide an estimate of this dispersion by building a generalized linear mixed model [25] in which random effects are described as smartphone-dependent and subject-dependent intercepts on top of the ordinary linear model defined above. Applying this analysis to the set of 436 calibrated values and allocating random intercepts to 218 subjects and 26 smartphone groups we can attribute 29% of the total variance to inter-subject variability, and 28% to smartphone model variability ($F_{(1,305)}$ = 311, $p < 0.001$).

## Discussion

### Analysis of comfort ratings

Situations of acoustical discomfort are way less frequent than situations of comfort, in our sample, which is of course desirable in a residential context. Note that, while the attribution to noisy or quiet states only labels the two extreme states at participants' locations, comfort ratings indicate participants' subjective perception of comfort. For a given location it's very likely that the quiet state will be rated more comfortable than the noisy state, however it may happen that the two extreme states are perceived as both comfortable or both uncomfortable. The examination of the asymmetry in the ratings in Fig 1 sheds light on this point.

While in quiet situations high comfort ratings are dominant (4 and 5 cover 84% of the data, 1 is never used), the ratings for noisy situations are much more variable: ratings 2, 3 and 4 are used with similar frequencies (25%, 36%, 22%), and even 5 (the most comfortable in the scale) is used in 4% of the cases (see Fig 1). This implies that noisy scenarios actually correspond to a variety of perceptions, not necessarily strongly associated with discomfort.

Measurements referring to quiet states are more frequently performed from rooms facing private spaces than public places open to traffic (56% vs 48%). However the mean comfort

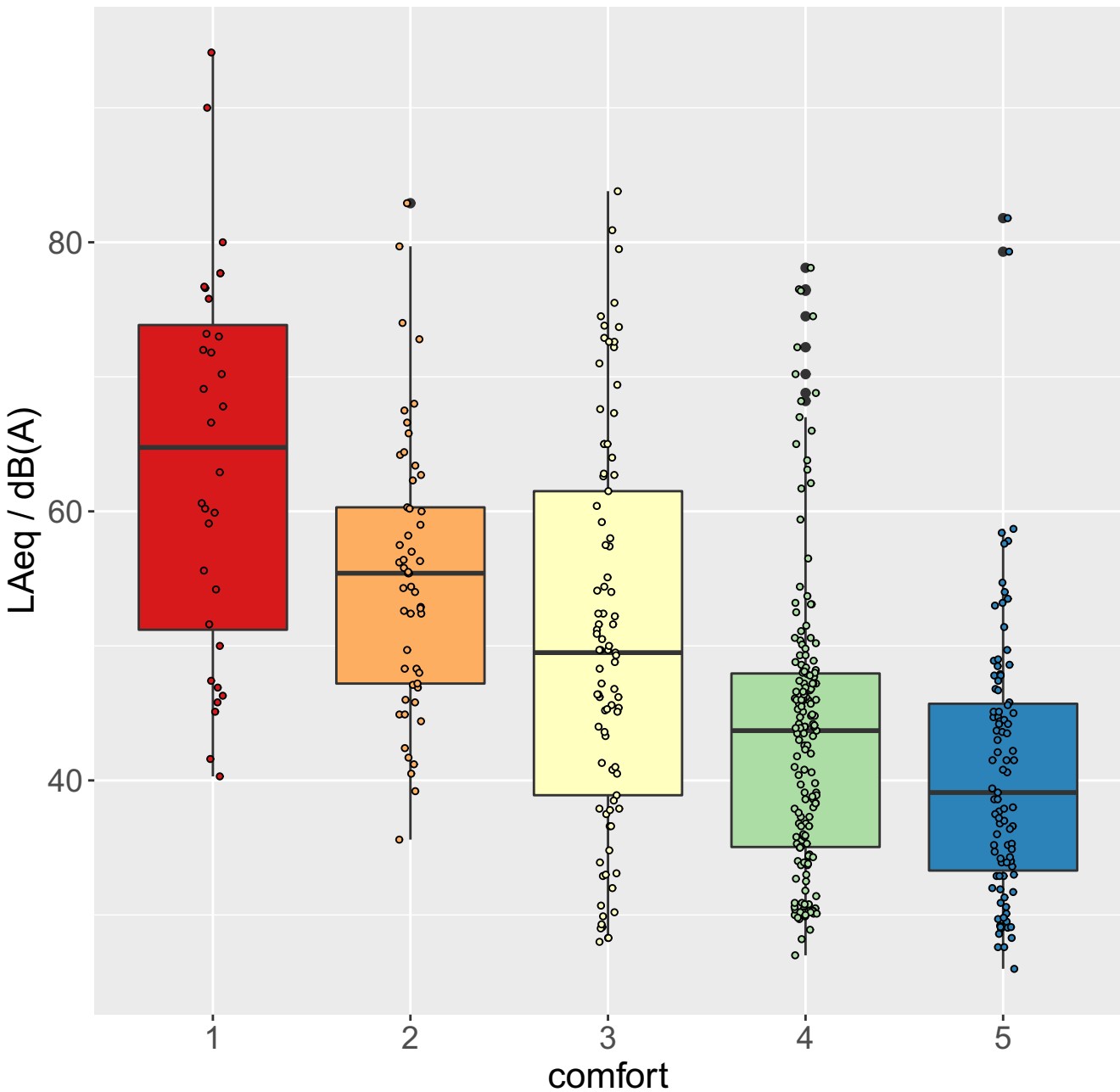

**Fig 5. Calibrated LAeq by comfort rating.** Thick black lines indicate the median values; the box limits the first and third quartile.

ratings are very similar (3.6 vs 3.5), probably indicating that participants considered inner-facing rooms of their homes *a priori* more representative of quiet states.

**The mean comfort scores in both quiet and noisy states appear to be decreasing with increasing population density.** However (see S3 Fig in S1 File), while the difference between mean comfort scores of isolated and highly populated places is always sizeable, places with a low population density are rated more acoustically comfortable than highly populated areas only in noisy situations (difference = 0.49), while they are almost equivalent in quiet situations (difference = 0.1). A more in-depth analysis about the sources of noise is therefore in order.

The distribution of the mean comfort scores by source (see S4 Fig in S1 File) shows a very interesting feature: despite some sources (such as air traffic) being reported too rarely to be relevant, the order of sources ranked by decreasing comfort is almost exactly the same in both low and quiet states, with natural sources being associated to the highest comfort and industrial sources to the lowest one.

Having found how sources are associated with mean comfort, we can examine how often each source is mentioned in a given population density environment. We find (see S5 Fig in S1 File) that the occurrence of road traffic noises increases monotonically with increasing population density. Natural sources show the opposite trend. Indoor anthropogenic sources are cited with similar frequency of occurrence in all environments. Noise from undefined sources is more frequently reported from low-density and isolated places in quiet situations.

This body of data clarifies the similarities noted above between low-density and high-density places in noisy states as they contain an equal fraction of road traffic. In quiet states, on the other hand, low-density places and isolated places share an equal fraction of natural or unidentified sources, which are overall rated among the most comfortable.

## Role of impulsive noise

Great care was taken in instructing participants about how to minimize extraneous noises originating from inside their own home, or due to smartphone mishandling, however, during a 1 min measurement, LAeq may be legitimately driven by sudden external events. This is reflected by Pearson's correlation between LAeq and LAmax, which is 0.94 in our dataset, while LAeq and LAmin correlation are 0.64 in noisy and 0.74 in quiet states. Exploiting this fact the occurrence of disturbing events can be inferred by studying the distribution of *LAeq— LAmin*. The spread of this distribution (see S6 Fig in S1 File) decreases monotonically with increasing comfort, confirming that disturbing events occur less frequently the quieter the environment is. Since LAmax and LAeq are strongly correlated, and *LAmax—LAmin* is an estimator of impulsive noise, these data show that the comfort rating is also directly affected by the annoyance of impulsive noise.

In contrast with measurements of Phase II (which are mostly louder than the quietest background) Pearson's correlation between LAeq and LAmin in Phase III is very high (0.88). This fact indicates that the measurements were overall much less affected by sudden disturbing events than in Phase II. However, even considering Apple smartphones only, it is clear from Fig 3 that about half of the background measurements were affected by some disturbance in Phase III, confirming that the measurement of a quiet background requires extra care to be performed. Nevertheless 75% of the values were < 41.4 *dB(A)*.

## Analysis of calibrated noise levels

In Fig 4 we reported that median gain-corrected LAeq are 49.1 *dB(A)* in noisy and 41.5 *dB(A)* quiet situations. Interestingly, these values are very close to the threshold values below which the impact of noise is considered negligible by Italian regulations. In fact, according to the Ministerial Decree setting threshold values for noise pollution [26], if the noise level measured indoors with open windows in residential areas does not exceed 40 *dB(A)* at night or 50 *dB(A)* during daytime, further analyses based on differential noise levels are not necessary.

Both distributions in Fig 4 have a peak centered at about 46 *dB(A)*. Around this level about half of the measurements are attributed to a quiet state and half to a noisy state. Noises with LAeq sound levels greater than this threshold are more likely to be classified as noisy than quiet. We interpret the ratio between noisy attribution to the total number of measurements, calculated in 3 *dB(A)* bins, as the average probability that environmental noise is rated as

annoying. Within a dynamic range in which calibrated smartphones appear to behave linearly (40–80 *dB(A)*) as *P[not quiet] / % = 1.08 × LAeq / dB(A)*, with 95% CI: [0.98,1.15] (see S7 Fig in S1 File). This relation only provides averaged information over all of the different types of noise experienced. It might also be thought as an estimation of annoyance, bearing in mind the *caveat* that subjective annoyance also depends on temporal and spectral features of sound as well as on day-time or night-time [27].

Actually the adequacy of the conventional 50/40 dB(A) day/night thresholds can be put to a test by the data we collected. In Fig 6 the cumulative fraction of records reporting a high level of comfort (4 or 5) are plotted as a function of the corresponding measured LAeq. The fractions corresponding to the nominal threshold are highlighted. During day-time 82% of the reports are below 50 *dB(A)*, and the curve shows a clear change in slope, confirming that this threshold is fully adequate. The fraction for night time is slightly lower (69%), and no pronounced change in the slope is visible at this value, suggesting that the night-time threshold could be more questionable, but the sample size at this time is smaller, producing larger uncertainty, so no certain conclusion can be drawn at this stage.

## Factors affecting noise levels and annoyance

S8 Fig in S1 File shows that the median LAeq at night (37.6 *dB(A)*) is 8.9 *dB(A)* lower than the daytime median LAeq. However, **median LAeq in quiet nights is 10.8 *dB(A)* lower than in noisy ones, and median LAeq in quiet daytime is 6.3 *dB(A)* lower than noisy daytime**. This decomposition shows that the feature at 30 *dB(A)* only appears on quiet nights while a large plateau of values between 30 and 50 *dB(A)* appears on quiet days. Identifying the sources of noise helps understanding these features.

The analysis of LAeq distribution by sources in 3 *dB(A)* bins (Fig 7) shows that road traffic explains about half of the peak around 46 *dB(A)* in both quiet and noisy states (see also S9 Fig in S1 File for a split version of this plot). The other half in quiet states mostly comes from natural sounds, while in noisy states it has mixed origins. Road traffic has a much larger level spread in noisy states than in quiet states. Together with outdoor anthropogenic sources it is the main contributor at all LAeq levels.

Natural noise is largely dominant in quiet states. Unidentified background noise is basically only present in quiet states, and is the main contribution to the peak at 30 *dB(A)*. To find the reason why noise from unidentified sources at low LAeq is so frequently reported we must examine how noise levels are distributed by population density.

Analyzing levels and sources by population density (S10 Fig in S1 File) clarifies two aspects. First, it turns out that the huge peak at 30 *dB(A)*, which we have determined to be characteristic of quiet nights, is only observed in low-density places. This is reasonable, since, in quiet nights, those places are most likely reached by noise not made locally (e.g. noise coming from faraway roads, industrial plants, human activities, etc.). A further explanation of this peak could be related to the intrinsic limits of most smartphones which, in practice, cannot measure lower levels. Second, it unravels the role of road traffic noise in different contexts.

Road traffic is present both in low and high-density areas, however the level distribution is quite different. A high volume of road traffic in low-density places produces lower median noise levels (46.2 *dB(A)*) than in high-density ones (51.1 *dB(A)*). On the contrary, a low volume of road traffic, in low-density places produces higher median noise levels (46.1 *dB(A)*) than in high-density ones (43.4 *dB(A)*). Outdoor anthropogenic noise shows a similar trend.

After Nov 11 some restrictions at the regional level were enforced by the authorities to contrast the COVID-19 pandemic. These measures were way less restrictive than the nation-wide lockdown enforced between Feb and Apr 2020. Essentially, mobility and activities were

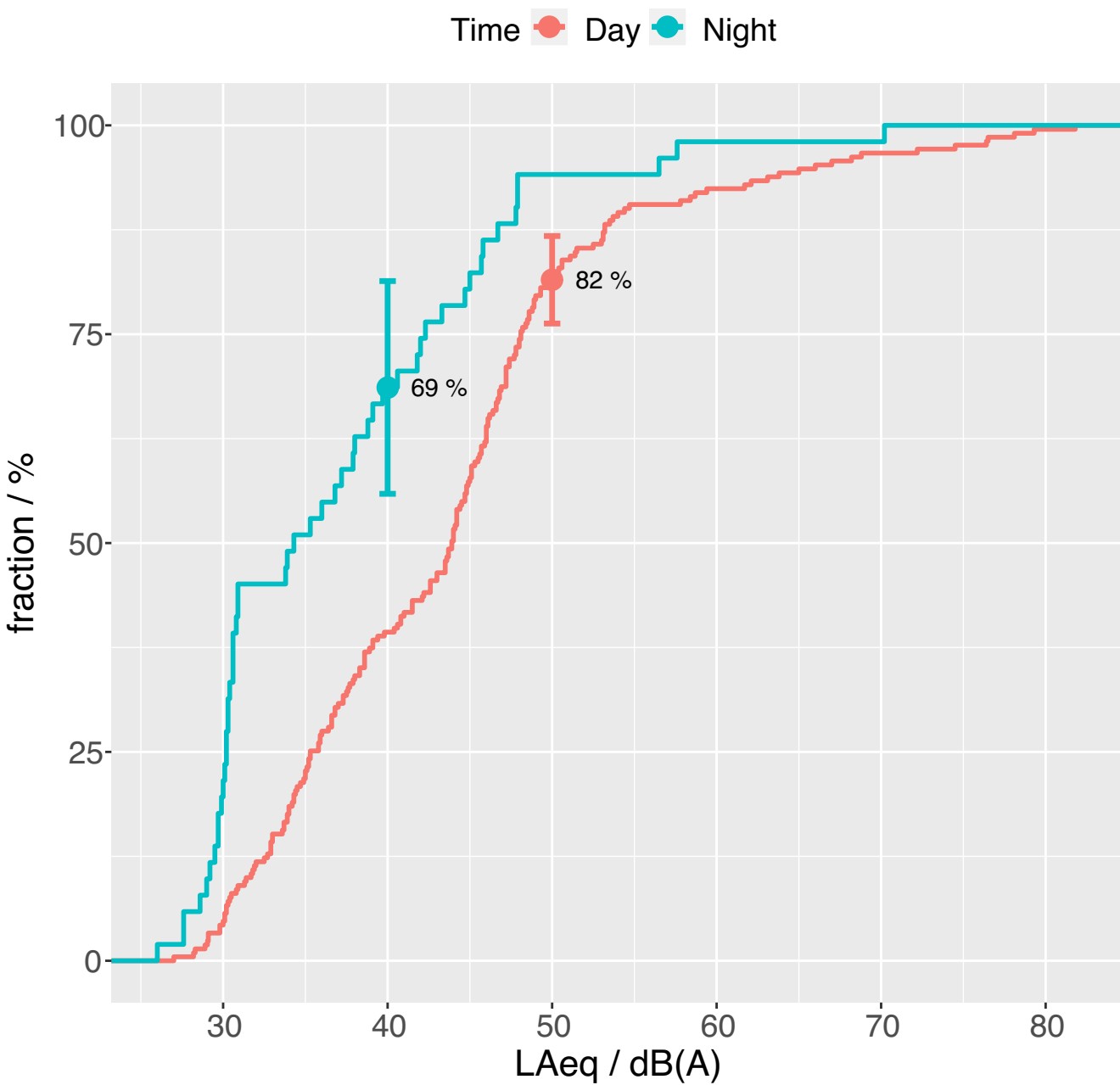

**Fig 6. Cumulative fraction of records reporting high comfort levels (4 or 5) as a function of the corresponding measured LAeq.** Respectively 82% and 69% of the records refer to sound levels below thresholds of annoyance as established by Italian regulations (i.e. 50 *dB(A)* during day-time and 40 *dB(A)* at night). Error bars indicate 95% confidence intervals, which are [76%, 87%] for day-time and [56%, 81%] for night-time data. Sample sizes are N = 211 (daytime) and N = 51 (nighttime).

prohibited later than 10 pm. Half of the measurements were performed after Nov 11. In this experiment we could not identify any significant change in the measured noise levels or in the reported comfort ratings before and after the restrictions were introduced. The reason is most probably that 89% of the measurements were performed during daytime, mostly in the after-noon, when no major restrictions to the circulation were effective (see S11 Fig in S1 File). Several studies report changes in LAeq [28, 29] and in the soundscape [30] associated with

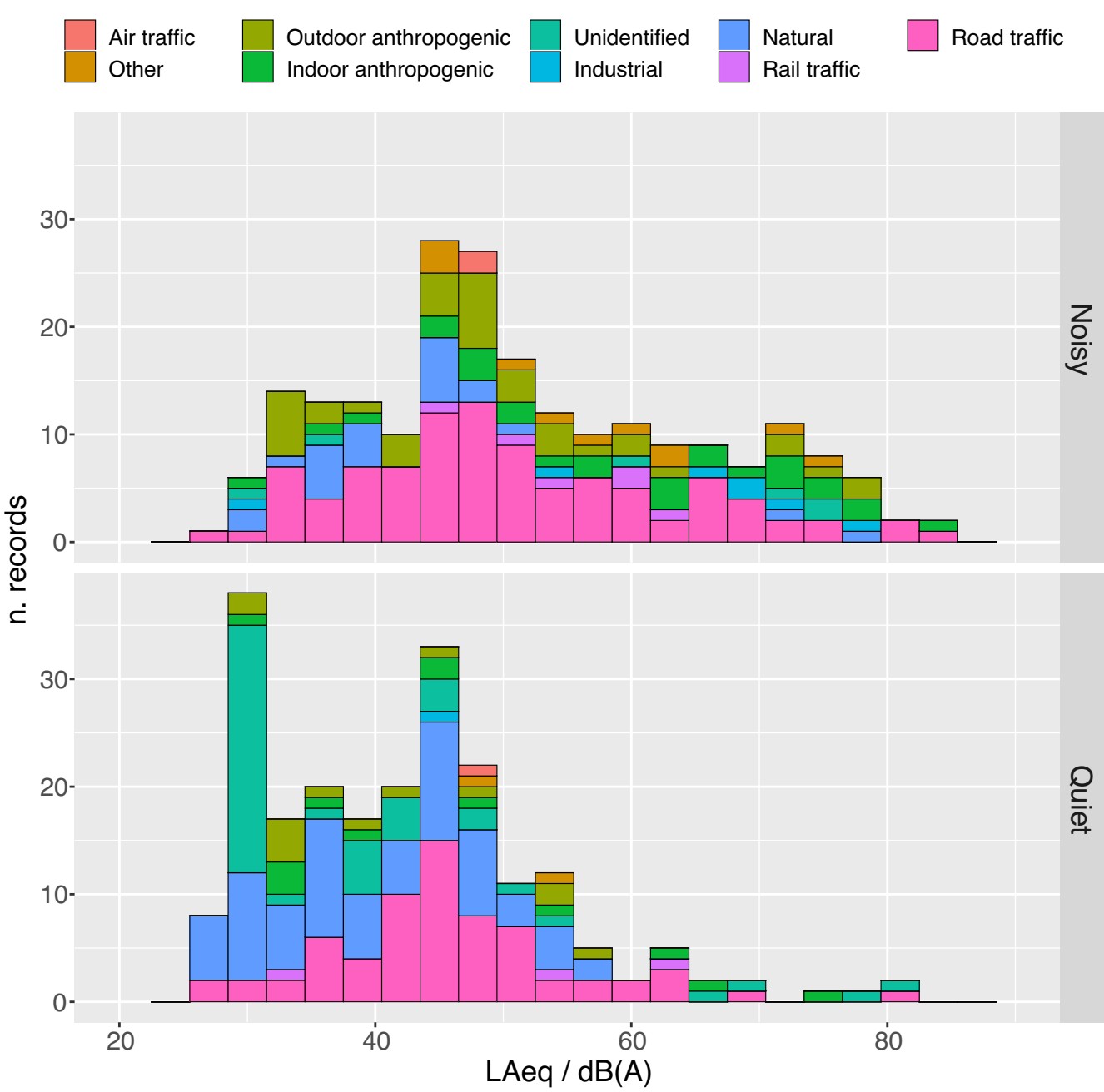

**Fig 7. Distribution of LAeq in low and high comfort states.** The color code indicates the sources in each 3 *dB(A)* bin.

COVID-19 restrictions to the circulation with variable impact depending on the nature of the urban area examined.

Our data do not show statistically significant differences in LAeq among different days of the week. Calibrated data per hour are too few to make any quantitative analysis possible.

Several (optional) comments were attached to the data records. 7 express general appreciation for the initiative. 10 comments just confirm that the time chosen to perform the

measurements is indeed representative of the nominal environment of the participant's home (7 confirm quiet states, 3 noisy states). 9 report anomalies in quiet states (typically sudden events such as ambulances or cars passing by). Many comments specify the main identified source of noise: 11 outdoor anthropogenic (barking dogs, bells, loud people outdoors), 5 road traffic, 3 indoor anthropogenic (TV or loud people indoors), 2 mixed sources (notably one indicates very loud birds singing on top of external voices), 1 rail traffic. 2 mention that the measurements were taken in a particularly quiet situation attributed to anti-Covid restrictions.

## Conclusions and perspectives

We have shown that collecting interesting data from a distributed network of citizens on a national scale is indeed a feasible task. We believe that initiatives such as this one, besides their societal impact, also have a potential for actual scientific data collection.

The campaign was mostly powered by students and teachers, for whom dedicated didactical materials were prepared. The consistency of the collected data was remarkably high, considered the heterogeneity of the contributions. We focused on spreading as much as possible the measurements in space, while, also based on previous experiences, we did not mean to perform a continuous time monitoring. As a consequence night-time measurements are a minority in our sample.

We have identified two main human-related difficulties and sources of errors. The first is the two-step process consisting in separately collecting data on paper first, and then manually selecting and sending them via a web interface. In this procedure the main mistakes concerned mistyped values and accidental swap of data belonging to quiet and noisy states. The simplification of this process, e.g. by allowing the app to directly upload data, would help limiting this kind of errors. The second source of outliers is due to some participants performing measurements in correspondence of extraordinary events, instead of restraining to situations more representative of their normal environment. Adding a dedicated question to the online form could help avoiding this problem.

Regarding gain corrections, we have shown that the analysis of differences between couples of uncalibrated values already provides a fair amount of information. By examining differential data only we could already unambiguously link the variation of noise levels to the variation of comfort ratings. However, if absolute levels are desired, gain corrections are necessary. In this case differential data can be employed as a further validation of gain adjusted values.

We have shown that applying a frequency-independent gain per smartphone model is a good starting point, the bottleneck being the continuous injection of new models on the market, which, in turn, requires that the calibration database is updated regularly if the experience is repeated over time. Few models for which the calibration was unreliable were easily spotted thanks to the measurement of the background noise.This indicates that Phase III actually is a key part of the protocol. From our data a scenario of moderate comfort appears also in situations considered noisy by the observers. The median sound levels are ordered increasingly from quiet-night to noisy-night, quiet-day, noisy-day. LAeq in quiet situations are less dispersed than in noisy situations, however in both cases few outliers appear due to sudden disturbing events. The body of calibrated LAeq values correlates well with the reported comfort ratings. Larger contributions of natural and undefined sound sources are found in quiet situations, and the comfort ratings follow a general trend, common to all situations, with natural sound rated the most comfortable and industrial sound the least comfortable. Different sources produce different distributions of LAeq, with characteristic shapes. All this information was employed to explain the overall shape of LAeq distributions. Differences between places having different population densities also emerge both in terms of LAeq and sources.

Despite a substantial amount of information can be extracted from the data, a large variability is to be expected, and is indeed observed. This fact confirms that, from the mere point of view of environmental acoustics, the main value of participatory science does not consist in the accuracy of the results. It rather rests on the opportunity to extract general trends by acquiring large datasets in the field in a cheap and quick way. As a perspective, the possibility of collecting spectral data is surely appealing, as it would definitely enable more in depth examinations of statistical sound level as annoyance indicators and the possibility of re-tuning the initiative in the direction of soundscape description, however the latter aspect is critically limited by the poor reliability of data collected by smartphones without the aid of external microphones at frequencies $< 250$ Hz. Finally experiments like the one we performed could also be employed to further select, among the participants, a subset of the most interested and active contributors, in order to organize a more focused campaign to refine the dataset, the general compromise being the optimal balance between the number of observations and the possibility of training individuals to strictly comply to standardized procedures.

## Supporting information

**S1 File. Contains all the supporting text, tables and figures.**
(DOCX)

## Acknowledgments

We thank the #scienzasulbalcone project group: Cecilia Tria and the social editorial staff of the Communication and Public Relations Unit of Cnr; Alessandro Farini and Elisabetta Baldanzi, Cnr–INO; Luca Perri, astrophysicist and scientific communicator; Fabio Chiarello, Cnr—IFN. We thank Giovanni Brambilla, Cnr—IDASC "Corbino", Massimo Nucci and Antonino Di Bella, Università di Padova for insightful discussions. We also thank the anonymous participants who joined the initiative and contributed data to this study.

## Author Contributions

**Conceptualization:** Carlo Andrea Rozzi, Francesco Frigerio, Luca Balletti, Silvia Mattoni.

**Formal analysis:** Carlo Andrea Rozzi, Francesco Frigerio, Daniele Grasso, Jacopo Fogola.

**Funding acquisition:** Carlo Andrea Rozzi.

**Methodology:** Carlo Andrea Rozzi, Francesco Frigerio.

**Software:** Carlo Andrea Rozzi, Luca Balletti, Daniele Grasso, Jacopo Fogola.

**Supervision:** Silvia Mattoni.

**Validation:** Daniele Grasso, Jacopo Fogola.

**Writing – original draft:** Carlo Andrea Rozzi.

**Writing – review & editing:** Carlo Andrea Rozzi, Francesco Frigerio, Luca Balletti, Silvia Mattoni, Daniele Grasso, Jacopo Fogola.

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
