## [Decision Letter · Decision Letter 0]

30 Sep 2021

PONE-D-21-26255Indoor noise level measurements and subjective comfort: a smartphone-based participatory experimentPLOS ONE

Dear Dr. Rozzi, 

Thank you for submitting your manuscript to PLOS ONE. After careful consideration, we feel that it has merit but does not fully meet PLOS ONE’s publication criteria as it currently stands. Therefore, we invite you to submit a revised version of the manuscript that addresses the points raised during the review process.

ACADEMIC EDITOR:The manuscript is addressing an important issue of indoor noise level measurement. There are some issues in methods and results section noted by the reviewer which needs further revision. 

We look forward to receiving your revised manuscript.

Kind regards,

Prashanth Prabhu

Academic Editor

PLOS ONE

2. Please update your Methods section to include the information about consent and data anonymization that you provided in the Ethics Statement.

Reviewers' comments:

Reviewer's Responses to Questions

**Comments to the Author**

1. Is the manuscript technically sound, and do the data support the conclusions?

Reviewer #1: Partly

2. Has the statistical analysis been performed appropriately and rigorously? 

Reviewer #1: Yes

3. Have the authors made all data underlying the findings in their manuscript fully available?

Reviewer #1: Yes

4. Is the manuscript presented in an intelligible fashion and written in standard English?

Reviewer #1: Yes

5. Review Comments to the Author

Reviewer #1: I thank the editorial board for providing me the opportunity to review this article, submitted to your reputed journal. There are considerable major limitations with the manuscript, which obscure the acceptance of the manuscript, in the current form. I recommend that the editor can review the concerns raised and request the authors to resubmit the manuscript, addressing the issues below.

Comments to the authors

TITLE: please consider title on the lines of ‘Feasibility of -------------- ‘.

INTRODUCTION: Rationale of the study needs to be strengthened. Many issues cited below needs to be addressed by the authors:

a) Although the authors discuss on various apps available for indoor noise measurements, and cite the limitations, it is very general. No specific data on reliability of their claims is available, specific to the apps cited. The authors have missed on the test-retest reliability, or accuracy of the cited apps in noise measurements.

b) The rationale on geospatial tracking breeching the privacy is not a strong reason, as the participants are asked to sign the informed consent on all the apps cited in introduction, before their usage. The terms and conditions of the spatial tracking would have been brought to their notice under disclaimer.

c) The DATA was collected in 2020, during the nation-wide lockdown of Italy. Hence, the noise levels in the inside house will be more (as more worked from home) and surrounding neighborhood activity would have considerably increased (say houses in the flats like residencies). The authors themselves point out this limitation in Introduction section (Page 9, lines 66-68) that the data was collected in Pandemic times. In my view, the data hence will not be representative of the day-to-day regular activity, which is the cause of concern for publication.

d) The data collected was emailed to the experimenters. The participants were asked to send the representative data of the nosiest and a quietest situation. This would induce bias and lead to perceptual judgement (if participants chose not to send a particular scenario based on the sound quality). Authors have not cited how this issue was controlled.

METHODS:

a) The reliability of the smart phones/ android devices in noise measurements is cited in general in introduction chapter. However, the qualitative differences in recording due to differences in handsets configurations across participants is not controlled. Additionally, the microphone sensitivity and its frequency resolution were not controlled across participants, which induces an inherent variability in data.

b) How was the distance aspect from the source in the room controlled. Eg. Say, the INVERTER being the highest noise generating object. In this case placing the smart phone at 30 cms from the source to 1 meter from the source will drastically affect the representative sample, although in both cases inverter generated noise may be the highest. Authors have not reported of any mechanisms to control such biases. Placing 1 m from the open window is the only condition mentioned, how abt the noise source. No information given.

c) Although the ambient noise was measured in few participants, the results of these findings or analyses corresponding to the same is not reported.

d) The participants were asked to read and record the values of LAmin, LAeq(t) and LAmax; based on which the subject would have chosen the sample to email. If the subjects were to use perceptual noise judgements to send the samples, choose which is noisiest and quietest, asking them to record the values would induce bias on their comfort ratings.

e) How did the authors ensure that the participants followed the instructions and the environmental influence on noise recording. Eg. incomplete opening of window, carpeting in the room, sound absorbers in room etc. A considerable variability can stem from them, which is not discussed.

RESULTS:

The authors need to support a lot of inconsistencies seen in the findings reported:

a) Lines 198-199: Different configurations of phones with different microphone sensitivity, and models can bring in a lot of variability in data and questionability about the reliability of the data collected. Authors state that ‘36 % of the smartphones employed were made by Apple running iOS, the remaining 64% were from mixed brands running Android (33% Samsung, 16% Huawei, 10% Xiaomi)’ resounds the above concern.

b) Lines 216 – 219: The authors state that : “In 70 cases (which contained valid data) the attribution to noisy and quiet states appeared to be inverted (i.e. both lower comfort rating and higher LAeq were attributed to the quiet state and vice versa). The labels for noisy and quiet states in these records were swapped”. This shows that the instructions were not correctly followed by the participants and this could be just one such instance. The data collected could be reflective of many such erroneous interpretations. The reasoning for inclusion and reversing the data, is not really satisfactory.

DISCUSSION: The authors discussion on the findings is elaborate and commendable. In my opinion although attempt has been made to categories the findings into different sub-headings, the readability is obscured due to the inclusion of results also into discussion. I suggest the authors rewrite this section with clear distinction of the results and only explain the findings in discussion, rather than putting in values and statistical analyses in this section. Two such instances are given below:

Lines 228-230: should be in results, not in included in discussion.

Line 462-463: Inferential statistics should be in results section

6. PLOS authors have the option to publish the peer review history of their article (what does this mean?). If published, this will include your full peer review and any attached files.

Reviewer #1: No

---

## [Author Response · Author response to Decision Letter 0]

9 Nov 2021

Please see the attached file "response to reviewers".

---

## [Decision Letter · Decision Letter 1]

6 Jan 2022

Indoor noise level measurements and subjective comfort: feasibility of  smartphone-based participatory experiments

PONE-D-21-26255R1

Dear Dr. Rozzi,

We’re pleased to inform you that your manuscript has been judged scientifically suitable for publication and will be formally accepted for publication once it meets all outstanding technical requirements.

Kind regards,

Prashanth Prabhu

Academic Editor

PLOS ONE

Additional Editor Comments (optional):

The authors have incorporated all the suggestions provided by the reviewer. The article may be accepted for publication.

Reviewers' comments:

Reviewer's Responses to Questions

**Comments to the Author**

1. If the authors have adequately addressed your comments raised in a previous round of review and you feel that this manuscript is now acceptable for publication, you may indicate that here to bypass the “Comments to the Author” section, enter your conflict of interest statement in the “Confidential to Editor” section, and submit your "Accept" recommendation.

Reviewer #1: All comments have been addressed

2. Is the manuscript technically sound, and do the data support the conclusions?

Reviewer #1: Yes

3. Has the statistical analysis been performed appropriately and rigorously? 

Reviewer #1: Yes

4. Have the authors made all data underlying the findings in their manuscript fully available?

Reviewer #1: Yes

5. Is the manuscript presented in an intelligible fashion and written in standard English?

Reviewer #1: Yes

6. Review Comments to the Author

Reviewer #1: (No Response)

7. PLOS authors have the option to publish the peer review history of their article (what does this mean?). If published, this will include your full peer review and any attached files.

Reviewer #1: **Yes: **K.V. NISHA

---

## [Editor Report · Acceptance letter]

10 Jan 2022

PONE-D-21-26255R1 

Indoor noise level measurements and subjective comfort: feasibility of  smartphone-based participatory experiments 

Dear Dr. Rozzi:

I'm pleased to inform you that your manuscript has been deemed suitable for publication in PLOS ONE. Congratulations! Your manuscript is now with our production department. 

Kind regards, 

on behalf of

Dr. Prashanth Prabhu 

Academic Editor

PLOS ONE